# Developmental phylotranscriptomics in grapevine suggests an ancestral role of somatic embryogenesis
Sara Koska[1,6], Dunja Leljak-Levanić [2,6 ✉], Nenad Malenica[2,6 ✉], Kian Bigović Villi[1,6], Momir Futo[1,3], Nina Čorak[1], Mateja Jagić [2], Ariana Ivanić[2], Anja Tušar[1], Niko Kasalo[1], Mirjana Domazet-Lošo[4], Kristian Vlahoviček [5] & Tomislav Domazet-Lošo [1,3 ✉]

The zygotic embryogenesis of Arabidopsis, which is initiated by gamete fusion, shows hourglass-shaped ontogeny-phylogeny correlations at the transcriptome level. However, many plants are capable of yielding a fully viable next generation by somatic embryogenesis—a comparable developmental process that usually starts with the embryogenic induction of a diploid somatic cell. To explore the correspondence between ontogeny and phylogeny in this alternative developmental route in plants, here we develop a highly efficient model of somatic embryogenesis in grapevine (*Vitis vinifera*) and sequence its developmental transcriptomes. By combining the evolutionary properties of grapevine genes with their expression values, recovered from early induction to the formation of juvenile plants, we find a strongly supported hourglass-shaped developmental trajectory. However, in contrast to zygotic embryogenesis in Arabidopsis, where the torpedo stage is the most evolutionarily inert, in the somatic embryogenesis of grapevine, the heart stage expresses the most evolutionarily conserved transcriptome. This represents a surprising finding because it suggests a better evolutionary system-level analogy between animal development and plant somatic embryogenesis than zygotic embryogenesis. We conclude that macroevolutionary logic is deeply hardwired in plant ontogeny and that somatic embryogenesis is likely a primordial embryogenic program in plants.

The ontogenies of multicellular eukaryotes are commonly marked by macroevolutionary imprints at the molecular level[1–8]. Curiously, we recently found that similar regularities are also present in the development of bacterial biofilms; a process that mimics embryogenesis of multicellular eukaryotes[9]. However, an hourglass-shaped correlation between ontogeny and phylogeny is a unique feature of multicellular eukaryotes. This pattern was first discovered in various animals by several independent approaches that compared the evolutionary conservation of genes and the ontogenetic timing of their expression[1–3]. Subsequently, the hourglass-shaped ontogeny-phylogeny correspondence was also discovered in the transcriptomes of a plant species *Arabidopsis thaliana*[4,6,7]. This was a largely unexpected finding because embryogenesis in plants, in contrast to animals, does not hint at the existence of such correlations at the morphological level[7,10]. However, until now, correlations between ontogeny and phylogeny in plants have only been

explored in the zygotic embryogenesis (ZE) of *A. thaliana*[4,6,7], leaving the existence of such correlations in other plant species or alternative developmental routes uncertain.

The organismal development of both animals and plants usually starts with the zygote formation and unfolds through the process of embryogenesis. In flowering plants, ZE involves double fertilization following the simultaneous formation of the embryo and the endosperm within a developing seed after a set of morphological, cellular, and molecular changes[11]. However, in contrast to animals that mainly have sexual ontogeny, life cycle in plants includes another level of complexity in the form of somatic embryogenesis (SE). During this process plant embryos develop from cells other than the fertilized eggs[12]. This is an alternative road to embryo-mediated plant formation, which typically includes reprogramming of somatic cells towards the embryogenic pathway, mostly after

[1]Laboratory of Evolutionary Genetics, Division of Molecular Biology, Ruđer Bošković Institute, Zagreb, Croatia. [2]Division of Molecular Biology, Department of Biology, Faculty of Science, University of Zagreb, Zagreb, Croatia. [3]School of Medicine, Catholic University of Croatia, Zagreb, Croatia. [4]Faculty of Electrical Engineering and Computing, University of Zagreb, Zagreb, Croatia. [5]Bioinformatics Group, Division of Molecular Biology, Department of Biology, Faculty of Science, University of Zagreb, Zagreb, Croatia. [6]These authors contributed equally: Sara Koska, Dunja Leljak-Levanić, Nenad Malenica, Kian Bigović Villi. ✉e-mail: dunja@zg.biol.pmf.hr; malenica@biol.pmf.hr; tdomazet@irb.hr

exogenous auxin treatment[13]. The best-known example of naturally occurring SE is the genus *Kalanchoe*, commonly called the mother of thousands, in which somatic embryos form spontaneously from diploid somatic cells on leaves edges[14].

In many plant species, SE can be artificially induced in different cell types after exposing the cells to various SE-promoting conditions[15]. Akin to a zygote, a dedifferentiated somatic cell that initiates SE usually shows cell polarity[16]. The subsequent development of somatic embryos, at least in *Arabidopsis* and other dicots, roughly follows morphological transitions characteristic for ZE; i.e., globular, heart, torpedo, and cotyledonary stages[17]. At the molecular level, some key developmental regulators, such as BBM and SERK1, are shown to be active both in somatic and ZE. Actually, some transcription factors like FUS3 and AGL15, which play a central role in ZE, are also involved in the ectopic embryo initiation of SE[18].

On the other hand, the currently available comparative transcriptome analyses of somatic and zygotic embryos, although limited to only a few developmental stages, reveal substantial transcriptional differences between these two developmental routes for a huge number of genes[19–21]. These large disparities between ZE and SE transcriptomes are perhaps not surprising given that many striking differences between zygotic and SE exist at the morphological and functional levels. For example, zygotic embryo development occurs inside the seed, where intensive communication between the embryo and surrounding endosperm occurs[11].

Similarly, zygotic embryos go through the phase of metabolic quiescence and desiccation, which is a part of seed maturation known as seed dormancy[22]. However, somatic embryos undergo the full developmental trajectory in the absence of these processes. These differences between zygotic and somatic embryogenesis likely alter developmental constraints and adaptive pressures that shape ontogeny-phylogeny correlations along these processes[1]. In SE, this could result in either an absence of correlation or a closer alignment with the theoretical hourglass profile compared to ZE[4]. Unlike ZE, SE can be triggered by a broader range of factors, including stressors such as tissue wounding or high concentrations of plant growth regulators like auxin[12,16,19]. All of this implies that SE, despite similar final developmental outcomes compared to ZE, is a unique developmental route in seed plants.

Although there are many studies encompassing transcriptomes of ZE from the pre-globular stages onward[7,19,21,23–25], very little is known about the molecular aspects in the very first steps of zygotic embryo development following fertilization[26] which is deeply embedded into maternal tissue and thus hardly accessible[27]. In this context, SE has a great advantage because somatic embryos are accessible at any stage of their development, which makes SE a favorable model of plant embryogenesis[28]. Another advantage of SE is that it allows an easy clonal propagation which is helpful in situations where efficient production of large numbers of genetically identical plants is required[29].

Several studies explored transcriptomes of somatic embryos in different plant species[19,20,30–34]. Unfortunately, these studies focus on a single or a few stages of SE, thus covering only a fraction of this developmental process[19,20,31–33]. In addition, RNA samples in some of these studies are derived from a mixture of different SE developmental stages[30–34], which precludes the recovery of time-resolved transcriptional trajectories. These limitations of currently available SE datasets make them unsuitable for studying phylogeny-ontogeny correlations, given that this type of analysis requires relatively dense sampling of individual stages along the full developmental process[35].

To overcome these limitations and to explore phylogeny-ontogeny correlations along the full SE developmental process, we established a highly efficient protocol for the direct induction of SE in grapevine (*Vitis vinifera* L.) "Malvasia Istriana", a perennial woody dicotyledon species and a cultivar with an international reputation. The developmental stages of our *V. vinifera* SE morphologically roughly resemble the stages of normal ZE, and the resulting embryos possess a high potential for immediate plantlet regeneration.

We used this SE system to sample 12 morphologically distinct developmental stages, covering the complete ontogeny of *V. vinifera* SE, from early induction to juvenile plant formation, and sequenced their transcriptomes using RNAseq. We matched the obtained expression values to gene-related evolutionary and functional information to explore the correspondence between ontogeny and phylogeny along the SE developmental trajectory. To achieve this, we applied phylostratigraphic and phylotranscriptomic methods which are very powerful in extracting macroevolutionary information from genomic and developmental data[1,4,9,36–40].

Here we show a strongly supported hourglass-shaped developmental trajectory in *V. vinifera* SE. Moreover, the recovered SE patterns better align with theoretical expectations than previously found profiles in ZE. This suggests that SE is a primordial process tightly linked to the evolutionary origin of development in plants.

## Results

### Global expression profiles along SE

To evolutionary assess developmental transcriptomes of SE, we first developed a highly efficient SE induction system in grapevine (*Vitis vinifera* L.) "Malvasia Istriana", which is characterized by a high embryogenic potential, developmental synchronization between embryos after induction, and the absence of fusion between embryos at their interfaces (see "Methods"). These properties of our SE system allowed us to relatively easily select and isolate individual embryos at different developmental stages in sufficient amounts for downstream RNAseq analysis (Fig. 1a). Embryogenesis was induced from immature anthers which are the most reactive explants for SE in different grapevine cultivars[41–43]. To cover the full embryogenesis as well as postembryonic development, we used three cultivation media and different lighting conditions that simulate embryo development and germination (Fig. 1a). This allowed us to gather a collection of 12 SE developmental stages covering induction, embryonic, and postembryonic development, including plantlet formation (Fig. 1a). To our knowledge, this is the most complete SE sample collection generated so far.

To get an overview of expression dynamics along SE in *V. vinifera*, we recovered the transcriptomes of these 12 SE stages by RNAseq (Fig. 1a). When considering all sequenced developmental stages together, we detected expression of 29,839 (99.56%) annotated *V. vinifera* genes (Supplementary Data 1). These high numbers showed that essentially all genes were transcribed at some point along the developmental trajectory of SE. In turn, this reveals that SE cannot be considered some simplified derivative of ZE, but rather a full-fledged developmental process that utilizes essentially all available protein-coding genetic information.

We further tested expression dynamics along the whole SE ontogeny which revealed that 25,098 (84.1%) genes were differentially expressed (Supplementary Data 2). Among these 12,893 (43.2%) had expression change above two-fold. To get a more detailed overview of expression dynamics we also compared transcriptomes in a pairwise manner between successive stages (Supplementary Data 2 and Supplementary Fig. 1). This pairwise analysis revealed that, on average, 18% of genes (12% with a fold change >2) showed changes in expression during transitions between successive stages. The most dramatic shift was observed during the transition from the C1 to C2 stage, where 37% of genes (25% with a fold change >2) exhibited altered expression (Supplementary Data 2 and Supplementary Fig. 1). These values, together with clustering analysis (Supplementary Data 3 and 4), revealed that SE in *V. vinifera* is a highly regulated process underpinned by substantial changes at the transcriptome level.

To get a global overview of the similarities and differences between the expressed transcriptomes of different developmental stages, we calculated pairwise expression correlations, which revealed that the SE developmental trajectory could be divided into five expression phases (Fig. 1a, b). The early expression phase includes the early induction (EI), pre-globular (PG), and globular (G1, G2) developmental stages (Fig. 1a, b). The mid-expression phase covers the heart (H) and early torpedo (T1) developmental stages. This mid-expression phase is followed by the late torpedo (T2) and early cotyledonary (C1) developmental stages which have rather unique transcriptomes that show some discontinuous similarity to the mid and late-expression phase (Fig. 1b). Finally, the late-expression phase comprises late

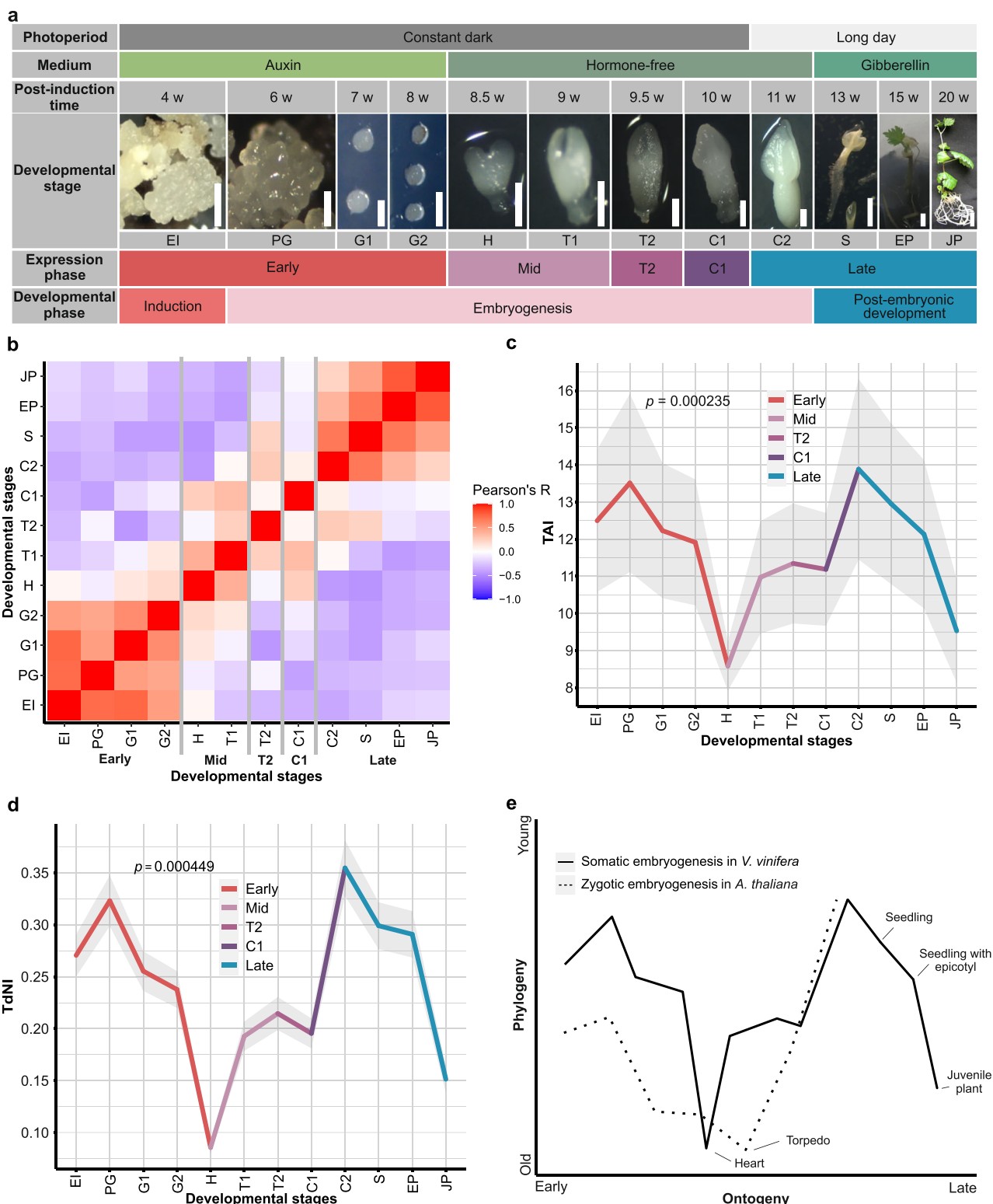

cotyledonary (C2), the formation of seedling (S), seedling with epicotyl (EP) and juvenile plant (JP) developmental stages (Fig. 1a, b).

To get further insights into expression dynamics along the SE developmental trajectory, we performed principal component analysis (PCA) which revealed a time-resolved profile that follows the developmental progression of SE and shows its punctuated organization (Fig. 2). The general organization of this PCA pattern in SE is similar to those previously recovered in bacterial biofilm development[9]. This suggests that these

developmental processes, although analogous, are governed by the common basic principles. Similar to bacterial biofilm development[9], biological replicates per developmental stage generally clustered together (Fig. 2). The only stage that showed increased distortion in expression between replicates is cotyledonary stage 1 (C1). This pattern in C1 could reflect a burst of expression changes, which potentially could be resolved in future studies by even finer temporal sampling around this specific period. Alternatively, this could point to an increased sensitivity of this particular stage to the slight

**Fig. 1 | Somatic embryogenesis in *V. vinifera* is a stage-organized process that exhibits an hourglass-shaped phylogeny-ontogeny correlation. a** The sampled developmental stages of somatic embryogenesis in *V. vinifera*: early induction (EI), pre-globular stage (PG), globular stage 1 (G1), globular stage 2 (G2), heart stage (H), torpedo stage 1 (T1), torpedo stage 2 (T2), cotyledonary stage 1 (C1), cotyledonary stage 2 (C2), seedling (S), seedling with epicotyl (EP) and juvenile plant (JP). Scale bars: 0.5 mm (EI–C2), 2 mm (S), 3 mm (EP), 1 cm (JP). The somatic embryogenesis stages were determined following previously described morphological criteria[43]. We performed transcriptome sequencing in $n = 5$ (C2 and EP stages) and $n = 3$ (the remaining 10 stages) biological replicates. For every sampled developmental stage, we showed corresponding hormones that were present in media as well as photoperiod at which developing plants were cultivated. "Long day" marks photoperiod of 18 h light and 6 h dark. For an easy reference, we also depicted post-induction time in weeks (w), global developmental phases, and expression phases derived from our correlation analysis. **b** Pearson's correlation coefficients between somatic embryogenesis developmental stages in all-against-all comparison. Early (EI–G2), mid (H–T1), T2, C1, and late (C2–JP) expression stages are marked. **c** The transcriptome age index (TAI) of somatic embryogenesis shows an hourglass pattern. The heart

stage of the mid-developmental period expresses the evolutionary oldest transcriptome, while earlier and later stages express evolutionary younger ones. We tested the significance of the TAI pattern using the flat-line test, while the gray shaded area represents ±1 standard deviation estimated using permutation analysis (see "Methods"). **d** The transcriptome nonsynonymous divergence index (TdNI) of somatic embryogenesis shows an hourglass pattern. The heart stage of the mid-developmental period expresses the most conserved genes at nonsynonymous divergence sites, while earlier and later stages express more diverged genes. Nonsynonymous divergence rates were estimated in *V. vinifera–V. arizonica* pairwise comparisons (see "Methods"). We tested the significance of the TdNI pattern using the flat-line test, while the gray shaded area represents ±1 standard deviation estimated using permutation analysis (see "Methods"). The corresponding transcriptome synonymous divergence index (TdSI) and transcriptome codon bias index (TCBI) profiles are shown in Supplementary Fig. 2e. **e** A schematic comparison between the TAI profile of *V. vinifera* somatic embryogenesis that we recovered in this study and the TAI profile of *A. thaliana* zygotic embryogenesis reported previously[4]. To make the hourglass patterns visually comparable between these studies, the TAI values were normalized to a range between 0 and 1 (see "Methods").

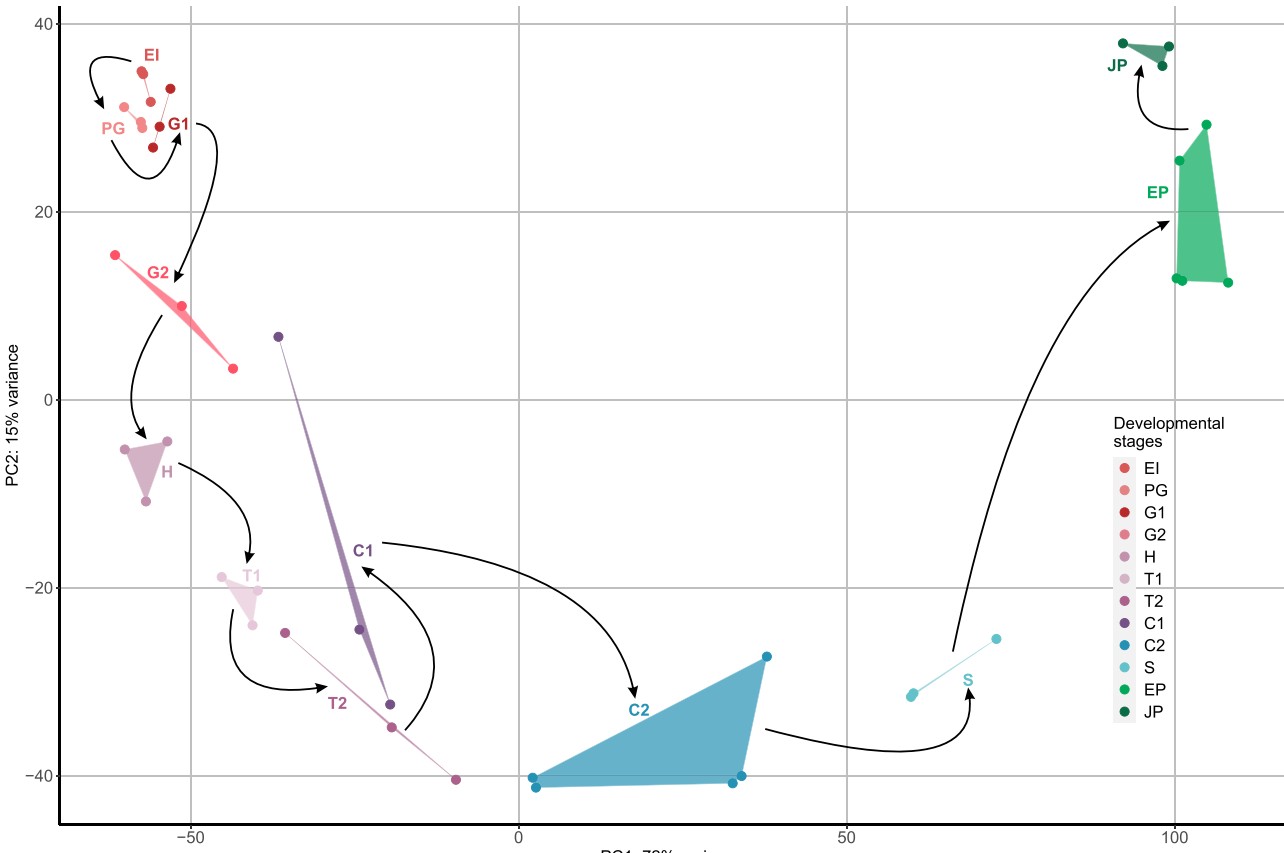

**Fig. 2 | Principal component analysis (PCA) of transcriptomes shows time-resolved organization of *Vitis vinifera* somatic embryogenesis.** *V. vinifera* developmental stages are shown in different colors, where shades of red represent early (EI–G2), shades of purple represent mid (H–T1), T2 and C1, and green represent the late developmental stages (C2–JP). Replicates are in the same color and connected with lines. We performed transcriptome sequencing in $n = 5$ (C2 and EP stages) and $n = 3$ (the remaining 10 stages) biological replicates. Black arrows correspond to the experimental timeline of *V. vinifera* development that starts with EI and ends at JP. Developmental stages: early induction (EI), pre-globular stage (PG), globular stage 1 (G1), globular stage 2 (G2), heart stage (H), torpedo stage 1 (T1), torpedo stage 2 (T2), cotyledonary stage 1 (C1), cotyledonary stage 2 (C2), seedling (S), seedling with epicotyl (EP) and juvenile plant (JP).

changes in environmental cues. We previously observed similar patterns during biofilm growth in developmental stages which were impacted by substantial environmental stress caused by starvation[9]. The fact that C1 stage is the latest stage kept in constant dark (Fig. 1a)—which causes a tradeoff between the lack of photosynthesis and developmental growth—suggests that higher variability in transcriptomes between biological replicates in C1 stage likely reflects the effect of starvation (Fig. 2).

## SE ontogeny-phylogeny correlations

To determine whether *V. vinifera* SE shows any correlation with the evolutionary trajectory of the plant lineage, we linked the transcriptome expression values of 12 SE developmental stages with the evolutionary age of *V. vinifera* genes and calculated the transcriptome age index (TAI)[1] (Fig. 1c; Supplementary Data 5). We assessed the evolutionary age of *V. vinifera* genes using a phylostratigraphic approach[9,36], based on a consensus phy-

logeny that traces back to the origin of cellular organisms and culminates in *V. vinifera* as the focal species, incorporating a large collection of reference genomes (see "Methods", Supplementary Fig. 3, Supplementary Data 6 and 7). We found that the TAI profile of *V. vinifera* development via SE has a pronounced, and statistically strongly supported, hourglass shape (Fig. 1c). The evolutionary younger transcriptomes are predominantly expressed during early development (early induction; EI and pre-globular stage; PG), after which increasingly older transcriptomes are recovered with the oldest estimates at the heart stage (H) (Fig. 1c). As the mid-development advances, evolutionary younger transcriptomes start to be expressed again, with the peak at cotyledonary stage 2 (C2), which showed overall the evolutionary youngest transcriptome (Fig. 1c). Finally, postembryonic developmental stages following the cotyledonary stage 2 (C2) exhibited a reverse trend, with transcriptomes becoming progressively older (Fig. 1c).

To test the stability of the recovered hourglass TAI profile we repeated the phylostratigraphic analysis using a range of *e* value cutoffs ($10-10^{-40}$) and recalculated TAI profiles[9,44]. This robustness test, which deliberately inflates false-positive and false-negative rates, demonstrated the stability of the hourglass TAI profile across the full range of tested *e* value cutoffs (Supplementary Fig. 4). This demonstrates that the TAI hourglass pattern of *V. vinifera* SE development is underpinned by a strong macroevolutionary imprint, which is resilient to the changes in *e* value thresholds. The strength of these macroevolutionary signals prompted us to look more closely at how different phylogenetic levels (phylostrata) contribute to the overall TAI profile. By sequentially including genes from successive phylostrata, starting from the ps1 (Cellular organisms), we recalculated a set of TAI profiles and found that the clearly recognizable, and statistically significant, hourglass pattern is detectable from the origin of Diaphoretickes (ps8) (Supplementary Fig. 5). These results suggest that the hourglass-shaped ontogeny-phylogeny correlations represent an ancient macroevolutionary imprint deeply embedded in the lineage that led to the origin of land plants.

To better understand the expression of genes from different phylostrata during *V. vinifera* SE, we conducted a relative expression analysis[1,9]. The genes that could be traced to the origin of cellular organisms (ps1) showed expression peaks at the heart stage (H) and in the juvenile plants (JP) (Supplementary Fig. 6). Similarly, genes that originated during archaeal diversification (ps2–ps5) and eukaryogenesis (ps6) also peaked around the heart stage (Supplementary Fig. 6). These expression peaks of evolutionary ancient genes at the heart stage (H) and in the juvenile plants (JP), explain in part why evolutionary oldest transcriptomes, as estimated by TAI analysis (Fig. 1c), are expressed at these stages.

With the exception of Diaphoretickes-specific genes (ps8) that show the highest expression at the cotyledonary 2 stage (C2), genes that emerged in the period from the origin of Excavata/Diaphoretickes (ps7) till the origin of Streptophyta (ps11) also showed maximal expression in the heart stage (H) (Supplementary Fig. 7a, b). This pattern demonstrates that genes that accompanied the early steps of the plant lineage diversification (ps7–ps11), which was unfolding in the aquatic environment, play an important role in the heart stage of extant SE. In contrast, the genes that originated from the origin of Embryophyta (ps12) to the origin of Magnoliophyta (ps14) showed very dynamic regulation across SE development (Supplementary Fig. 7c). Although genes from these evolutionary periods also have relatively high expression levels at the hearth stage (H), we detected strong additional peaks at the globular (G1 and G2), cotyledonary (C1 and C2), torpedo (T1 and T2), and seedling (S) stages (Supplementary Fig. 7c). Together this pattern showed that the genes that originated during the early evolution of land plants (ps12–ps14) play an important role in the period from the globular stages (G1) to the seedling (S) stage, i.e., the central part of SE ontogeny (Fig. 1a, Supplementary Fig. 7c).

The evolutionary young genes that originated in the period from the origin of Eudicots (ps15) to the origin of focal species *V. vinifera* (ps18) cumulatively follow the hourglass pattern (Supplementary Fig. 7d). Their upregulation is evident at the beginning of SE, in the early induction (EI) and pre-globular (PG) stages, as well as at the final phase of embryo maturation and during germination; i.e., in the cotyledonary 2 (C2) and seedling (S) stages (Supplementary Fig. 7). Taken together, the SE developmental hourglass is

underpinned by the upregulation of evolutionary older genes (Cellular organisms, ps1 to Spermatophyta, ps11) at the heart stage (H), and by the upregulation of evolutionary younger genes (Eudicots, ps15 to *V. vinifera*, ps18) at the beginning and the end of SE (Supplementary Figs. 6 and 7).

The TAI analysis relies on the evolutionary origin of unique sequences in the protein sequence space; hence it reflects a deep macroevolutionary history. However, to answer the question of whether the hourglass profile is maintained in the more recent evolutionary periods, we estimated the divergence rates between orthologous coding sequences of *V. vinifera* and *V. arizonica* (Fig. 1d, Supplementary Fig. 2a and Supplementary Data 5) and linked these values with SE expression trajectories. This approach originally used the ratio between nonsynonymous and synonymous substitution rates (dN/dS ratio) to calculate the transcriptome divergence index, assuming that synonymous substitution rates are a proxy of neutral evolution[4]. However, synonymous substitutions cannot be considered neutral when selection acts on synonymous sites, e.g., via the codon usage bias[9]. To account for this effect, we previously devised transcriptome nonsynonymous divergence index (TdNI) and transcriptome synonymous divergence index (TdSI), which allowed us to independently study how divergence rates at nonsynonymous and synonymous sites correlate with expression levels[9].

We found that both the transcriptome nonsynonymous divergence index (TdNI) and the transcriptome synonymous divergence index (TdSI) in *V. vinifera*–*V. arizonica* comparison show a clear and statistically supported hourglass profile (Fig. 1d, Supplementary Fig. 2a and Supplementary Data 5). The genes with the lowest divergence rates, at both nonsynonymous and synonymous sites, are predominantly expressed at the heart stage (H). In contrast, the genes with the highest divergence rates are prevailingly expressed in the induction (IE) and pre-globular (PG) stages, at the onset of SE development, and in the cotyledonary 2 (C2) stage, at the end of embryogenesis (Fig. 1d, Supplementary Fig. 2a and Supplementary Data 5). Interestingly, TdNI and TdSI curves closely follow the TAI pattern, suggesting that similar forces operate at different evolutionary scales. Additionally, transcriptome codon bias index (TCBI) showed that in *V. vinifera* SE genes which are expressed during the heart stage (H) and in juvenile plants exhibit the strongest codon usage bias (Supplementary Fig. 2b and Supplementary Data 5). Altogether, these results confirm the existence of an hourglass-shaped ontogeny-phylogeny correlation in SE development in relatively recent evolutionary history that spans *V. vinifera*–*V. arizonica* divergence.

The TAI profile that we detected in the SE of *V. vinifera* could be tentatively compared to the one previously found in the ZE of *A. thaliana*[4] (Fig. 1e), in the part that covers embryogenesis *sensu stricto*, i.e., from the early induction (EI) to the cotyledonary 2 stage (C2). These profiles have rather similar shape (Fig. 1e), with a notable difference in that the SE of *V. vinifera* expresses evolutionary oldest genes at the heart stage, while the ZE of *A. thaliana* exhibits evolutionary oldest transcriptome at the subsequent torpedo stage[4]. Although TAI patterns for some parts of ZE postembryonic development of *A. thaliana* are available[7], it is unreliable to compare them to the SE postembryonic development of *V. vinifera* because the sampled stages in these studies do not match (Fig. 1e). For example, some ZE stages such as "mature dry seeds", "imbibed seeds", "seeds at *testa rupture*" and "radicle protrusion"[7], simply do not exist as a part of the SE seedless development. Nevertheless, similar to our study, this previous work also reports the existence of phylogeny-ontogeny correlations in postembryonic ZE development of *A. thaliana*[7]. Interestingly, the postembryonic drop in TAI values that we detected in the SE of *V. vinifera* (Fig. 1c), highly resembles the pattern of postembryonic development in animals which also shows a progressive drop in TAI values[1].

## Functional trends

To test the functional grouping of upregulated genes in specific developmental stages, we performed the enrichment analysis of GO functional categories (plant subset) across SE development. We found that every stage of SE has a specific battery of enriched GO functions (Fig. 3, Supplementary

**Fig. 3 | Functional enrichments along the somatic embryogenesis of *V. vinifera*.** We analyzed the enrichments of GO functional categories in genes that are upregulated in the different stages of somatic embryogenesis. A gene was considered upregulated in a particular stage if it was transcribed 0.5 times (log₂ scale) above the median of its overall transcription profile. The frequency of a GO annotation per stage is compared to the frequency of that annotation in the whole *V. vinifera* genome and shown as log odds (bubble graph). The log odds higher than zero denote that the frequency of annotation in a given developmental stage is higher than the expected frequency estimated from the whole genome. The significance of these functional enrichments was tested by a two-tailed hypergeometric test. The *p* values were adjusted for multiple testing (see "Methods"). Only significant enrichments are shown. The color code follows expression phases in Fig. 1a: early development (red), mid-development (violet), and late development (turquoise).

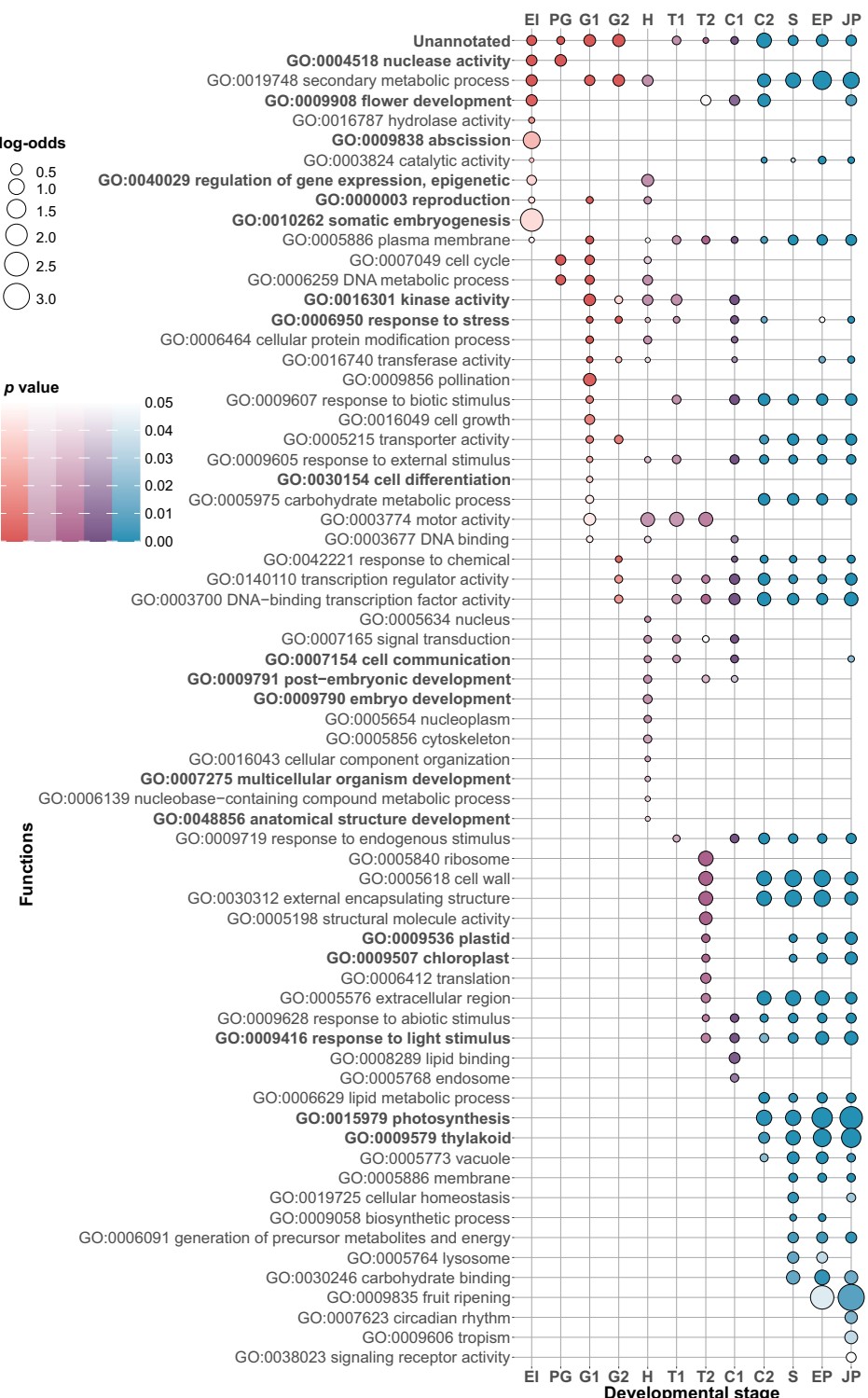

Data 8), which indicates that functional transitions along SE rely on extensive transcriptional regulation. Genes with unknown functions are enriched in all SE stages (Fig. 3, Supplementary Data 8), except in the heart stage (H). This pattern is congruent with the fact that the heart stage expresses evolutionary the oldest transcriptomes (Fig. 1c), and that functionally older genes are more often functionally studied (Supplementary Fig. 8). On the other hand, it is striking that many unannotated genes have regulated expression along SE (Fig. 3) and that most of them emerged during the diversification of land plants (ps12–ps18, Embryophyta to *Vitis vinifera;* Supplementary Data 8 and Supplementary Fig. 8). This shows that

our understanding of how embryonic development of land plants has evolved is markedly incomplete.

It is rather reassuring that the GO term "somatic embryogenesis" (GO:0010262) was strongly and significantly enriched at the early induction (EI) stage, which marks the onset of SE (Fig. 3, Supplementary Data 8). However, to get a deeper understanding of this functional enrichment, we plotted individual expression trajectories of six genes that contribute to this signal (Fig. 4a). Interestingly, five of them show a clear trend with the highest expression in the early induction (EI) and pre-globular (PG) stages, followed by increasingly lower expression levels toward juvenile plant (JP) stage

(Fig. 4a, Supplementary Data 4 and 9). Some of these genes, such as *AGL15*, *FUS3*, and *IAA30*, have *A. thaliana* homologs which are known to be important in promoting SE[12]. Furthermore, we observed significant

enrichment of GO terms related to metabolism, biosynthesis, and photosynthesis during late embryonic and postembryonic development (Fig. 3, Supplementary Data 8). This finding aligns with expectations, as protein

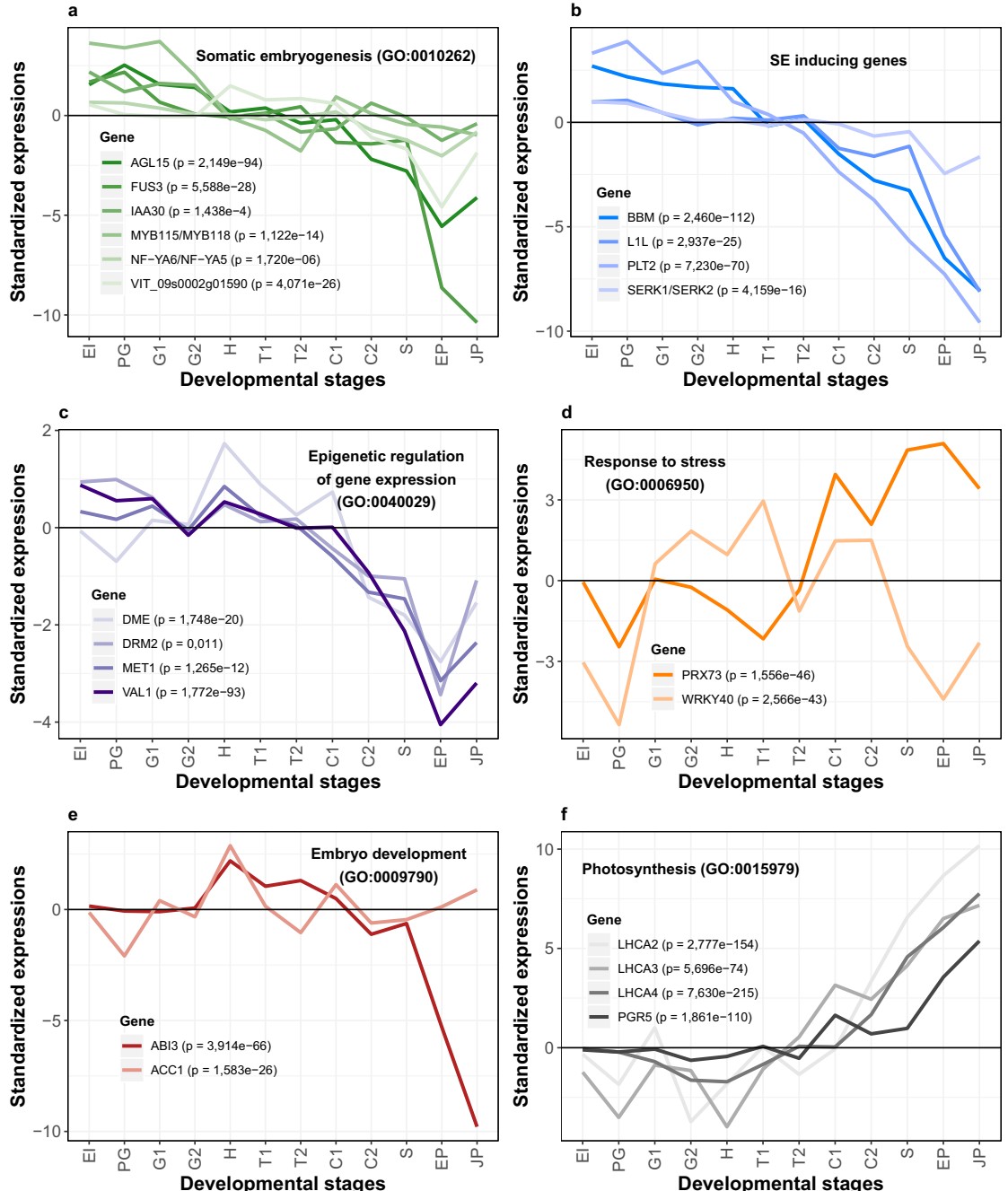

**Fig. 4 | Standardized expression profiles of selected *V. vinifera* genes along somatic embryogenesis. a** Standardized expression profiles of genes annotated with GO term "Somatic embryogenesis" (GO:0010262) that showed an enrichment signal in Fig. 3 (EI stage). **b** Selected genes that have an important role in the induction of somatic embryogenesis according to the literature. **c** Three representative genes (*DME*, *DRM2*, and *MET1*) annotated with GO term "Epigenetic regulation of gene expression" (GO:0040029). This GO term showed an enrichment signal in Fig. 3 (EI and H stages). *VAL1* is described in the literature to be important for chromatin modification. **d** Two representative genes (*PRX73*, *WRKY40*) annotated with GO term "Response to stress" (GO:0006950). This GO term showed enrichment signals in Fig. 3 (G1, G2, H, T1, and C1 stages). **e** Two representative genes (*ABI3*, *ACC1*) annotated with GO terms "Embryo development" (GO:0009790), "Multicellular organism development" (GO:0007275) and "Anatomical structure development"

(GO:0048856). These GO terms showed enrichment signals in Fig. 3 (H stage). **f** Four representative genes (*LHCA2*, *LHCA3*, *LHCA4*, *PGR5*) annotated with GO terms "Plastid" (GO:0009536), " Chloroplast" (GO:0009507), "Response to light stimulus" (GO:0009416), "Photosynthesis" (GO:0015979) and "Thylakoid" (GO:0009579). These GO terms showed enrichment signals in late developmental stages (C2, S, EP, and JP) in Fig. 3. Differential expressions along SE were tested by LRT test as implemented in *DESeq2*[91]. Resulting *p* values corrected by FDR are shown for every gene. Standardized expression value of 0 (black horizontal line) represents the median of expression levels for a respective gene. Gene names were obtained by searching for *V. vinifera–a. thaliana* orthologs in TAIR database[95]. Correspondence between *V. vinifera* and *A. thaliana* genes together with standardized gene expression values can be found in Supplementary Data 4, and standardized gene expression profiles in Supplementary Data 9.

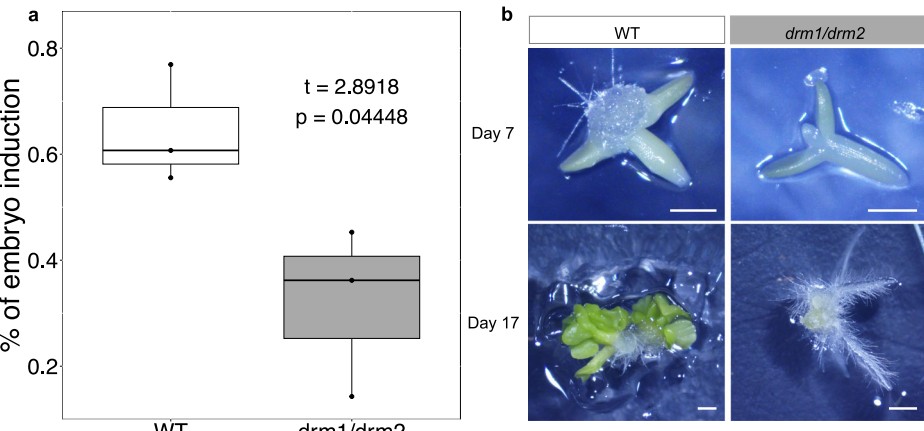

**Fig. 5 | The potential for embryo induction is significantly reduced in the *A. thaliana* drm1/drm2 double mutant. a** Embryo induction potential of wild type *A. thaliana* (WT) and *drm1/drm2* double mutant. The mean values of *n* = 3 individual experiments, each including 50 to 100 embryos per line, are shown (Supplementary Data 10). *P* value (Student's *t*-test) shows statistically significant differences between the mean values (95% confidence interval: 0.013–0.64, df = 4). The effect size was determined by calculating Hedges' *g*, which equals 1.88 and denotes a large effect size. Dots represent the actual percentage of embryo induction in each of the three individual experiments in each line. **b** *A. thaliana* somatic embryogenesis induction. Zygotic embryo explants of WT exposed to 2,4-D develop callus-like tissue between the cotyledons (WT, day 7). Somatic embryos formed 10 days after transfer to 2,4-D-free medium (WT, day 17). The absence of somatic embryogenesis induction in the *drm1/drm2* mutant line (*drm1/drm2*, day 7) resulted in the root proliferation and formation of non-embryogenic callus (*drm1/drm2*, day 17). Scale bar = 1 mm.

synthesis in a developing organism demands substantial energy investment[45]. This requirement is especially pronounced in plants, which, unlike animals, autonomously synthesize energetically costly amino acids[45].

Although GO annotation datasets are rather useful in screening general functional tendencies, they are nevertheless incomplete when it comes to the precise functional annotation of individual genes. We thus plotted the expression trajectories of additional genes which are known from the literature to play an important role in SE but lack this type of annotation in our GO dataset (Fig. 4b). Similar to GO-derived analysis, we found that all *V. vinifera* homologs of SE-important *A. thaliana* genes, such as *BBM*, *L1L*, *PLT2* and *SERK*[12], showed high expression at the onset of SE followed by increasingly lower expression levels toward the juvenile plant (JP) stage (Fig. 4b, Supplementary Data 4 and 9). This rather regular expression profile of many important SE genes qualifies them as useful markers for tracking SE in future studies, e.g., in single-cell RNAseq experiments.

Epigenetic regulation of gene expression plays an important role during phase transitions in the life cycles of plants[12,46,47]. In our analysis, we detected two significant enrichments for the GO term "Epigenetic regulation of gene expression" (GO:0040029), which suggests that the early induction (EI) stage and the heart (H) stage are especially important transition phases for epigenome reprograming in the SE of *V. vinifera* (Fig. 3). Because many genes (Supplementary Data 8) contribute to these enrichments, we illustrated general trends by depicting expression profiles for four representative epigenetic regulators (Fig. 4c). For example, methylase DRM2, which is responsible for de novo methylation[46], showed increased gene expression in the early induction (EI) stage as well as the heart (H) stage (Fig. 4c, Supplementary Data 4 and 9). We found a similar pattern for VAL1 (Fig. 4c, Supplementary Data 4 and 9) which is a transcriptional repressor involved in histone methylation[12,48]. On the other hand, DNA methylase MET1, required for the maintenance of DNA methylation during replication, and DNA demethylase DME[46] showed the highest gene expression in the heart (H) stage (Fig. 4c, Supplementary Data 4 and 9).

Reports on different plant species indicate that auxin-mediated induction of SE is linked to the activity of methylation-maintaining methyltransferases, such as MET1[49], while the functional loss of DRM-class de novo methyltransferases (DRM1 and DRM2) primarily affects gametophyte development[50]. Furthermore, the loss of function of DRM2 (since *DRM1* is not expressed in plant embryos) somewhat affects the patterning of cell division in the early zygotic embryo, likely related to the methylation patterns established in the egg cell[51,52]. Surprisingly, we found a high expression level of *DRM2* during *V. vinifera* SE, which even exceeds the expression level of *MET1* in the early SE phase (Fig. 4c).

To experimentally verify the significance of increased expression of a DRM-class methyltransferase in *V. vinifera* SE, we used the *A. thaliana* model system. The auxin presence/absence protocol for SE induction and maturation in *A. thaliana*[53] (see "Methods"), was performed on the wild type and the *drm1/drm2* double mutant[54]. The *drm1/drm2* double mutant exhibited a significantly lower SE induction potential (32%) compared to the wild type (64%) (Fig. 5a). Nevertheless, the successfully induced explants of the mutant line retained the capacity for full embryo maturation similar to wild type (Fig. 5b). These results demonstrate that DRM-class enzymes indeed impact the competence of explants for SE induction in *A. thaliana* and most likely also in *V. vinifera*, indicating egg cell-like behavior of SE-induced somatic cell.

It was suggested that the regulation of stress response plays an important role during SE because various stress-related genes have elevated expression in somatic embryos[16,19,31,34]. Our GO function enrichment analysis revealed that the GO term "Response to stress" (GO:0006950) is indeed significantly enriched in many stages over SE ontogeny including G1, G2, H, T1, C1, C2, EP, and JP stage (Fig. 3, Supplementary Data 8). Interestingly, the GO term "Abscission" (GO:0009908), which also could be linked to stress responses, is strongly enriched at the early induction (EI) stage (Fig. 3, Supplementary Data 8). The full list of genes which contribute to the enrichment of these terms and their profiles is available in Supplementary Data 8 and 9. As an example, we depicted WRKY40, which is a transcriptional repressor that functions in plant responses to pathogens and abiotic stresses within complex regulatory networks that include other *WRKY* genes[55]. *WRKY40* showed high expression in the middle period of *V. vinifera* SE, from the globular stage 1 (G1) to the torpedo stage 1 (T1) (Fig. 4d, Supplementary Data 4 and 9). In contrast, peroxidase *PRX73*, another stress-related gene, was highly expressed during postembryonic development including seedling (S), seedling with epicotyl (EP), and juvenile plant (JP) stages (Fig. 4d, Supplementary Data 4 and 9), where it likely has a role in controlling root hair growth by modulating cell wall properties[56].

Of all considered developmental stages, the heart (H) stage showed the most unique functional profile with several enriched GO functional categories related to embryo development (Fig. 3, Supplementary Data 8). This suggests that at the functional level, the heart stage is a critical period for embryonic development where the expressions of key embryogenic genes converge. Again, these functional enrichments were underpinned by many

genes (Supplementary Data 8 and 9). To illustrate major trends, we thus sorted out two examples (Fig. 4e). *ABI3*, one of the central regulators that initiate maturation in the heart stage of *Arabidopsis* ZE[57], showed a peak of expression in the heart stage (Fig. 4e, Supplementary Data 4 and 17). Similarly, a multifunctional enzyme ACC1, which is known for its role in cotyledon morphogenesis in the heart (H) stage of zygotic embryos[58], also showed maximal gene expression at the heart (H) stage of SE (Fig. 4e, Supplementary Data 4 and 9).

The late developmental stages (C2 to JP) showed functional enrichments related to photosynthesis such as "Plastid" (GO:0009536), "Chloroplast" (GO:0009507), "Thylakoid" (GO:0009579), "Photosynthesis" (GO:0015979) and "Response to light stimulus" (GO:0009416) (Fig. 3; Supplementary Data 8). This period corresponds to the switch from growth in the constant dark to a "long day" regime (Fig. 1a), hence one might expect the activation of photosynthesis-related genes. To show common expression trends of these genes, we depicted *LHCA* and *PGR5* genes as examples (Fig. 4f). *LHCA* genes encode for thylakoid light-harvesting chlorophyll-binding proteins that have a vital role in photosystem I[59], while PGR5 is essential for photoprotection and cyclin electron transport around photosystem I, especially in acclimation to fluctuating environments[60,61]. All of these photosynthesis-related genes showed a common trend where their expression values continuously increase during postembryonic development (Fig. 4f, Supplementary Data 4 and 9).

## Discussion

SE as an experimental model has the advantage over ZE because it enables the production of genetically identical somatic embryos in large numbers. On the other hand, the unsynchronized development of somatic embryos, as well as their aggregation into physically compact clusters represent the main obstacles of current SE protocols. Both problems limit the isolation of individual developmental stages without embryo wounding. For example, despite its huge advantages as a model system, *Arabidopsis* somatic embryos are fused along their contact surfaces. This leads to the formation of unsynchronized embryo clusters that cannot be easily separated without tissue damage[62,63]. Another limitation of *Arabidopsis* SE is the low proportion of embryos that complete embryogenic development, which consequently leads to the low frequency of plantlet regeneration. This limitation is further provoked by the culturing of somatic embryos for a long time[64]. To address these issues, here we developed a comparatively rapid protocol with a rather low input of growth regulators that enables synchronized development of unfused individual embryos with high plantlet regeneration potential. We view our *Vitis vinifera* "Malvasia Istriana" SE induction system as a highly reproducible and potentially widely applicable model for plant SE research. This potential is demonstrated by our discovery of an increased transcriptional profile for *DRM2* in the early stages of *V. vinifera* SE and by the observed loss-of-function phenotype in *A. thaliana* SE (Fig. 5).

Plant embryogenesis is an old process that most likely emerged at the root of Embryophyta (ps12), predating the later invention of seeds and flowers in seed plants[65–67]. In this context, ZE in flowering plants could be considered an evolutionary-derived process, which includes innovations such as endosperm formation, desiccation, and dormancy[66,68]. In contrast, SE, which does not depend on these adaptations, might be a better representation of the ancestral embryogenic trajectory of land plants.

Moreover, SE is not limited to seed plants, as this process also exists in ferns, which seem to show higher potential for SE induction than spermatophytes[69]. This indicates that probably all clades of Embryophyta retained the potential for SE. Taken together, the hourglass pattern that we discovered in the SE of *V. vinifera* is most likely a better proxy of ancestral phylogeny-ontogeny correspondence that underpinned Embryophyta diversification, than the one described in the ZE of *Arabidopsis*[4]. Interestingly, the phylotranscriptomics of brown algae, which lack canonical embryogenesis, shows conserved transcriptomes in the multicellular stages, while unicellular stages evolve more rapidly[8]. This suggests that, at least in brown algae, transcriptome conservation at particular stages is a broader

phenomenon associated with cell differentiation and not necessarily linked to embryogenesis[8,70,71].

The original study that discovered hourglass-shaped correlations between phylogeny and ontogeny in *Arabidopsis* ZE predicted that the phylotypic stage (the waist of hourglass) in plants should be placed somewhere between the globular and the heart stage[4]. This prediction assumes that a phylotypic stage should possess all major body parts at their final anatomical position in the form of undifferentiated cell aggregates[4]. However, this study detected a disparity between this prediction and the recovered phylotranscriptomic profile that shows the waist at the subsequent torpedo stage—a developmental stage which marks the beginning of the maturation phase linked to seed formation[4]. Obviously, this discrepancy between the theoretical predictions and the actual pattern is not present in the SE of *V. vinifera* where the waist of the hourglass phylotranscriptomic profile is placed at the heart stage, as originally expected. This finding suggests that SE, besides many technical advantages[66], is a better model to study general developmental principles in land plants than ZE.

From this perspective, SE could be viewed as an atavistic trait[72]. Although in some plants, like in some species of the genus *Kalanchoe*, SE is an integral part of the life cycle, in many others it can only be activated upon stress induction[73]. It seems that atavistic characters in plants are generally induced by the impact of stress[72], which occasionally pushes plant cells to the expression of ancient developmental programs[16,19,72]. In some instances, the co-option of atavistic programs obviously has an adaptive value. A good example is found in constitutive plantlet-forming species within the genus *Kalanchoe*, where the co-option of SE into leaves compensates for the propagation deficiency in *Kalanchoe* species that otherwise produce only nonviable seeds[73]. However, whether the stress induction of SE in natural settings has a broader prevalence and adaptive value remains unclear.

Plants and animals are intrinsically multicellular organisms that independently evolved their multicellularity[39,74]. Yet, their embryogenesis shows remarkable system-level analogy in the form of hourglass-shaped phylogeny-ontogeny correlations[4,10] and in the macroevolutionary dynamics of genome complexity change[39]. A recent study also reported the existence of a transcriptomic hourglass pattern in brown algae, which evolved multicellular development independently from animals and plants[8,71]. This suggests that the hourglass pattern might represent a convergent feature of complex multicellularity across distinct evolutionary lineages.

Our findings reveal that analogies in phylogeny-ontogeny correlations are particularly pronounced when comparing animal development to SE in plants. This similarity primarily relates to the positioning of the phylotypic stage in the mid-embryogenesis where the primordia of all major body parts are placed at their final anatomical positions[14]. However, it is also indicative that we found comparable trends in later phases of ontogeny. Namely, with the formation of seedling (S), we observed that *V. vinifera* plantlets increasingly express older and less diverged transcriptomes. This strongly resembles the pattern in animals where aging adult animals express increasingly older genes[1].

However, it remains unclear which evolutionary forces govern these analogies. In animals, several studies have attempted to elucidate the evolutionary mechanisms that maintain the developmental hourglass[75–78]. The full picture has not been revealed yet, but it seems that a combination of purifying selection[76] linked to pleiotropic effects at mid-embryogenesis[75] and positive selection acting on the early and late phases of embryogenesis[77] shape the hourglass profile in animals. Similar studies in plants are currently lacking; however, there is a possibility that evolutionary mechanisms behind developmental hourglass in plants are more complex than in animals. Namely, a recent study found that mutations occur less frequently in functionally constrained *A. thaliana* genome regions[79]. If this finding stands the test of time[80,81], this would open the possibility that developmental hourglass in plants, in addition to purifying and positive selection, is underpinned by mutational bias. In any case, the difference in the relative

positioning of the hourglass waist between zygotic and SE observed in this study suggests that the phylotypic stage may undergo heterochronic shifts.

In sum, we conclude that macroevolutionary imprint, in the form of hourglass-shaped ontogeny-phylogeny correlations, is deeply hardwired in plant ontogeny and is largely resilient to alternative developmental routes, such as zygotic and SE. Our discovery that the shape of ontogeny-phylogeny correlations in SE better fits with theoretical expectations and that it more closely resembles analogous patterns in animals, suggests that SE is likely a primordial embryogenic program in plants.

## Methods
### Plant material
Inflorescences of *Vitis vinifera* L. "Malvasia Istriana" (Malvazija istarska) were acquired from the National Collection of Autochthonous Grape Varieties of the University of Zagreb, Faculty of Agriculture experimental station "Jazbina" during the May/June, which was ~2–3 weeks before anthesis. Alternatively, we induced flowering by placing the basal part of dormant vine cuttings (~30 cm in length) into distilled water and by exposing them to 24 °C and 16/8 photoperiod using daylight florescent tube (40 W, 400–700 nm, 17 W/m²). Anthers were isolated from the buds of sterilized inflorescences according to the procedure described in Malenica et al.[43]. The *drm1/drm2* transgenic seeds with mutated methyltransferases DRM1 and DRM2[54] were ordered from NASC (The Nottingham Arabidopsis Stock Centre, donor: Steve Jacobsen, NASC ID N16383).

### Induction of embryogenesis
Modified MS medium[82], lacking glycine and with MS-nitrogen sources substituted with X6 nitrogen sources[83], were used as a basic medium in this study. Induction medium was prepared as the basic medium with the addition of 5 µM BAP (6-benzyladenine), 2.5 µM 2,4-D (2,4-dichlorophenoxyacetic acid), 2.5 µM NOA (naphthoxyacetic acid)[41], sucrose (2% w/v) and agar (7% w/v). The pH of the media was adjusted to 5.8 before sterilization at 121 °C, 103 kPa for 15 min.

Whole flower buds were aseptically removed from the inflorescence and opened by cutting the basal side of the bud. Filaments were excised at their bases using a medical needle under the stereomicroscope and, together with attached anthers, placed with their adaxial side facing the surface of the medium. Between 20 and 25 explants were cultivated in a 30 × 10 mm Petri dish at 24 °C in the dark.

### Embryo maturation
Globular embryos were transferred separately onto the hormone-free basic medium suitable for somatic embryo development, with the addition of 0.5 g/L activated charcoal[83]. Cultures were cultivated at 24 °C in the dark.

### Somatic embryo germination and plant regeneration
Cotyledonary stage embryos developed on hormone-free basic medium were induced to germinate on the embryo germination medium (EG) supplemented with 10 µM IAA and 1 µM GA3[84]. Cultures were exposed to 24 °C and 16/8 photoperiod using daylight florescent tube (40 W, 400–700 nm, 17 W/m²). The details of this procedure are described in Malenica et al.[43].

### Selection of different developmental stages of somatic embryos
Classification and selection of each specific developmental stage during and post-embryogenesis were based on morphological criteria for seven *Vitis vinifera* cultivars (ref. 43; Fig. 1a). The yellowish proembryogenic masses (EI) were formed on the filament tip. To collect single cells and cell clusters, the proembryogenic masses were mechanically separated from the filament and cultivated in liquid induction medium for 16 h with constant agitation in the growth chamber at 24 °C in the dark. After sieving the cell suspension through a 150 µm nylon mesh, cells and small cell clusters from the liquid phase were collected on 50 µm nylon mesh and split into two portions. One half was recultured for testing the embryogenic competence (induction success), while the rest was shock-frozen in liquid nitrogen and stored at −80 °C until further use. Only if recultured cells were efficient in embryo

production (pre-globular and globular embryo formation within 2 weeks), the corresponding frozen sample was used further.

Pre-globular (PG) and globular stage (G1 and G2, different in size) embryos were isolated from the embryogenic tissue by sieving the tissue through the metal mesh to remove the older stages and large clusters. The filtrate fraction that contained mostly PG and G embryos was washed further with a fresh liquid basic medium by using a 150 µm nylon mesh to remove single cells and small clusters.

The PG embryos were distinguished from the G stage according to the morphology of the epidermal cell layer. In contrast to well-formed epidermis of discrete globular stage embryos, the pre-globular stage was mainly attached to the explant tissue. In the cases when they were detached from the explant tissue we detected them by their surface which was not smooth and even (Fig. 1a). After collecting each stage separately, they were again re-washed with a basic medium using a 150 µm nylon mesh.

Later embryogenic stages (heart H, torpedo T1 and T2, cotyledonary C1 and C2) were isolated using fine forceps and a needle, based on their specific shapes and sizes observed under a binocular microscope. Collected embryos were washed with basic medium by using a 150 µm nylon mesh to remove the remaining tissue. The stages of postembryonic development were determined according to the development of root hairs, epicotyl, and the first pair of leaves.

### Somatic embryo induction and maturation in *Arabidopsis thaliana*
Immature zygotic embryos of the wild type and the *drm1/drm2* mutant line were used as explants for the induction of SE, which was performed according to Gaj[62]. Siliques containing cotyledonary embryos were collected, surface sterilized (1% NaOCl, 0.1% Mucasol™ solution), rinsed with sterile distilled water, and opened with insulin syringes. The seeds were carefully scraped into the drop of sterile distilled water and the embryos were carefully ejected with a coverslip. Fifty to one hundred cotyledonary zygotic embryos per line were planted onto SE induction medium (E5 + 2,4-D[62]). Explants were cultured for seven days under long-day conditions (16 h light/8 h dark, 120 µmol m⁻² s⁻¹) at 24 °C. After 7 days of cultivation on induction media, the induction potential of each line was calculated. The induced explants were transferred to maturation media (E5 without 2,4-D) and cultured for another 10 days under the same conditions to allow somatic embryos to develop.

### RNA extraction
Total RNA was isolated from somatic embryos using the RNeasy Plant Mini kit (Qiagen, Hilden, Germany) with slight modification of the manufacturer's protocol. Depending on the developmental stage, samples contained between 50 and 70 individual embryos. Each sample was homogenized in 450 µL RLT buffer with 20 mg PVP (polyvinylpyrrolidone; Sigma, St. Louis, USA) in a 2 mL plastic tube using four stainless steel beads (3 mm diameter). The bead beater (Retsch MM200, Haan, Germany) was set to 30 Hz for 3 min. The homogenate was filtered in a Qiashreader column at 10,000 rpm for 1 min. The supernatant was mixed with 0.5 volumes of 96% EtOH, transferred to an RNA binding column, and centrifuged according to the manufacturer's instructions. DNA removal was performed by applying 80 µL of DNase I working solution (10 µL DNase stock + 70 µL 1x RDD buffer; RNase-Free DNase Set, Qiagen, Hilden, Germany) to the column and incubated for 15 min at room temperature. Then 350 µL of RW1 buffer was added to the column and centrifuged at 10,000 rpm for 15 s. This washing step was repeated one more time. Further, two washing steps were performed with 500 µL of RPE buffer at 10,000 rpm for 15 s. The RNA was eluted with 40 µL of Tris-HCl pH 6.8 (Ambion, Austin, USA). Finally, 1 µL of RNase inhibitor (40 U/µL; Thermo Scientific, Waltham, USA) was added to the sample and incubated for 5 min at room temperature and stored at −20 °C before sequencing.

Isolated RNA was quantified on a Nanodrop spectrophotometer (Thermo Fischer, Waltham, USA). The A260/280 values for all samples were between 1.8 and 2.0 and the RNA concentrations were in the range of

20–190 ng/μL, depending of the embryo developmental stage. The RNA quality was tested by 1% agarose gel electrophoresis (1xTAE).

## RNA sequencing

Total RNA extracted from each somatic embryo developmental stage (EI to JP) was sent to the EMBL Genomics Core Facility (Heidelberg, Germany) for quality check, rRNA depletion, cDNA library preparation, and high throughput sequencing. The samples were sequenced in five (C2 and EP stages) and tree replicates (the remaining 10 stages) using the Illumina NextSeq 500 platform (read length 75 bp, paired-end). Sequence quality and read coverage were checked using the FastQC V0.11.9[85] with a satisfactory outcome for each of the samples. In total, 3,945,355,234 paired-end sequences (75 bp) were mapped onto the *V. vinifera* reference genome (NCBI Assembly Accession: 12X, GCA_000003745.2) using BBMap V38.75[86] with an average of 95.48% of mapped reads per sample (Supplementary Data 1). On average, we mapped 90 million reads per replicate (Supplementary Data 1). Mapping was performed using the standard settings with the option of trimming the read names after the first white space was enabled. Generating, sorting, and indexing of BAM files was done by using SAMtools V1.11[87]. These files were then used for the downstream data analyses in R V4.0.4[88] using custom-made scripts. Briefly, quantification of mapped reads for each *V. vinifera* open reading frame was done using the R *rsamtools* package V2.10.0[89] and raw counts for 29,839 (out of 29,971) open reading frames were retrieved using the *GenomicAlignments* R package V1.30.0[90]. We estimated expression similarity between replicates and developmental stages using the PCA analysis (Fig. 2) implemented in the R package *DESeq2* V1.34.0[91]. The obtained results were visualized in the R package *ggplot2* V3.3.5[92].

## Transcriptome analysis

To prepare the raw count values for the subsequent analysis, we normalized them by calculating the fraction of transcripts ($\tau$)[93]. The reasoning behind using $\tau$ for the downstream calculation of evolutionary measures has been discussed in previous work[1,9,94]. We resolved the replicates by calculating the replicate median for each developmental stage. The obtained normalized transcript expression values were used to calculate evolutionary indices (Supplementary Data 5), and relative expression values of phylostrata.

Following a pipeline introduced in previous work[9], we calculated the standardized expression values of each gene for use in GO enrichment analysis (Fig. 3), clustering (Supplementary Data 3 and 4), and profile visualization (Fig. 4, Supplementary Data 4 and 9). Briefly, we discarded genes that had the expression value of zero in more than two developmental stages, removing 3790 genes from the dataset. For genes that had a single stage with the expression value of zero, we interpolated it with the mean of the two adjacent stages (1064 genes), or if the expression value of zero was in the first or last stage, we transferred the value of the only neighboring stage directly (390 genes). Lastly, the expression values for each gene were normalized to the median and $\log_2$ transformed, resulting in the standardized expression values for 26,181 genes.

The standardized expression profiles were visualized (Fig. 4, Supplementary Data 3 and 4) using the R package *ggplot2* V3.3.3[92]. Genes selected for expression profile visualization in Fig. 4 were selected based on their GO annotations (Fig. 4a, c–f) and orthology to SE-relevant *A. thaliana* genes[12] (Fig. 4b, c). Gene names and gene orthologs between *A. thaliana* and *V. vinifera* were selected based on the TAIR database[95].

To cluster a large dataset with 26,181 genes, we first split the dataset into 13 randomly sampled groups of genes; 12 groups consisted of 2015 genes and one of 2014 genes. Using the DP_GP_cluster[96] with the maximum Gibbs sampling iterations set to 500, we clustered the standardized expression profiles of genes within each of the 13 groups which yielded 1157 gene clusters in total. For each of these clusters, we calculated the mean standardized expression profile. Using again the DP_GP_cluster with the maximum Gibbs sampling iterations set to 500 we clustered these 1157 mean standardized expression profiles, which finally resulted in the 85 clusters composed of 26,181 genes (Supplementary Data 3 and 4).

We tested the transcriptome similarity between different developmental stages by calculating Pearson's correlation coefficients (*R*) using standardized expression values for all-against-all comparisons and visualizing it on a heatmap (Fig. 1b). Using a pipeline implemented in the *DESeq2* V1.30.1[91] R package, we estimated the pairwise differential gene expression between the individual developmental stages (Supplementary Data 2 and Supplementary Fig. 1), as well as overall differential expression for every gene across all developmental stages (Supplementary Data 2) with the likelihood ratio test (LRT) implemented in the same package.

## Functional enrichment analysis

Due to the lack of a comprehensive set of functional gene annotations, we used eggNOG-mapper V2.0[97] to annotate the *V. vinifera* genome. We obtained the best annotation data using the default search filters and limiting the taxonomic scope to Eukaryota. This resulted in 18,425 genes annotated with GO annotations[98] (Supplementary Data 8). We then performed the functional enrichment of individual developmental stages using the assigned GO annotations. To simplify analyses, we limited GO terms used in the functional enrichment to the GO Plant subset downloaded from the Gene Ontology Resource website (GO version: https://doi.org/10.5281/zenodo.4735677, May 20, 2021). In addition, we included in this GO Plant subset the missing term GO:0010262 "Somatic embryogenesis" because it was relevant for our research. We tested the enrichment of these GO terms in each developmental stage for a set of genes that had in that particular stage a standardized expression value of at least 0.5 ($\log_2$ scale) above the median of their overall expression profile across SE (Fig. 3, Supplementary Data 4). All enrichment analyses were performed using the two-way hypergeometric test (Supplementary Data 8). To adjust for multiple comparisons, we corrected the *p* values using the Benjamini and Hochberg procedure[99].

## Evolutionary measures

The phylostratigraphic procedure was performed as described in previous work[36,37]. Following the latest phylogenetic literature[100,101], we constructed a consensus phylogeny covering the divergence from the last common ancestor of all cellular organisms to *V. vinifera* as the focal organism (Supplementary Fig. 3, Supplementary Data 6 and 7). Phylogenetic trees were visualized and annotated in the iTOL v6 online tool[102] (Supplementary Fig. 3 and Supplementary Data 6). The choice of internodes (phylostrata) in the consensus phylogeny depended on their phylogenetic support in the literature, the availability of reference genomes for the terminal taxa, and their importance for evolutionary transitions.

We retrieved the full set of protein sequences for 427 terminal taxa, five from NCBI and 422 from the Ensembl database (Supplementary Data 7). We prepared the referent protein sequence database for sequence similarity searches by checking the files for any inconsistencies, adding taxon tags to the sequence headers of all sequences, and leaving only the longest splicing variant of each eukaryotic gene. The phylostratigraphic map of *V. vinifera* was constructed by comparing 29,927 *V. vinifera* protein sequences against the referent protein sequence database using blastp algorithm V2.9.0[103] with the *e* value threshold of $10^{-3}$. Discarding all protein sequences which did not return a significant match left us with 29,623 protein sequences in the sample. We then mapped those 29,623 protein sequences on the 18 internodes (phylostrata) of the consensus phylogeny (Supplementary Data 7). Each protein sequence was assigned to the oldest phylostratum where it still had a blast hit[36,37].

For each developmental stage, using the expression values of 29,623 protein-coding genes, we calculated the TAI (Fig. 1c, Supplementary Data 5):

$$TAI = \frac{\sum_{i=1}^{n} ps_i e_i}{\sum_{i=1}^{n} e_i}$$

where $ps_i$ is an integer that represents the phylostratum of the protein *i*, $e_i$ is the normalized expression value of the gene *i*, and *n* is the total number of genes analyzed. Previous work discussed the biological interpretation of TAI and its statistical properties at length[1].

To compare *V. vinifera* SE TAI profile to *A. thaliana* ZE TAI profiles (Fig. 1e), we downloaded previously obtained *A. thaliana* ZE TAI values[4]. The TAI values from SE and ZE were normalized as follows:

$$Norm(TAI_s) = \frac{x_s - \min(x)}{\max(x) - \min(x)}$$

where $Norm(TAI_s)$ represents the normalized TAI value for the developmental stage s, $x_s$ is the TAI value of a developmental stage s, while $\min(x)$ and $\max(x)$ are the minimum and maximum TAI values across all developmental stages. The obtained normalized TAI values were plotted on the Y axis in a range from 0 (lowest TAI value) and 1 (highest TAI value) (Fig. 1e).

To test the robustness of the TAI profile and the phylostratigraphic pipeline in general, we used the blastp algorithm V2.9.0[103] to construct additional phylostratigraphic maps with different e value cutoffs (10, 1, $10^{-1}$, $10^{-2}$, $10^{-3}$, $10^{-5}$, $10^{-10}$, $10^{-15}$, $10^{-20}$, $10^{-30}$, $10^{-40}$) (Supplementary Data 7 and Supplementary Fig. 5). To calculate the divergence rates of *V. vinifera* proteins, we used the pipeline available in the R package *orthologr* V0.4.0[6]. Using blastp reciprocal best hits with $10^{-5}$ e value threshold, we found 18,761 orthologs in *Vitis arizonica* (Grape genomics: Vitis arizonica cl. b40-14 V1.1, https://doi.org/10.5281/zenodo.3827985) (Supplementary Data 5). After globally aligning *V. vinifera*–*V. arizonica* ortholog pairs using the Needleman-Wunsch algorithm, we used pal2nal to construct codon alignments[104]. We calculated the nonsynonymous substitution rates (dN), the synonymous substitution rates (dS), and the sequence divergence rates (dN/dS) using Comeron's method[105].

For each developmental stage, using the dN values of 18,751 genes, we calculated the transcriptome nonsynonymous divergence index (TdNI) (Fig. 1e, Supplementary Data 5):

$$TdNI = \frac{\sum_{i=1}^{n} dN_i e_i}{\sum_{i=1}^{n} e_i}$$

where $dN_i$ is a real number that represents the nonsynonymous divergence of gene i, $e_i$ is the normalized transcript expression value of the gene i, and n is the total number of genes analyzed[9]. For each developmental stage, using the dS values of 18,727 genes, we calculated the transcriptome synonymous divergence index (TdSI) (Supplementary Data 5 and 14):

$$TdSI = \frac{\sum_{i=1}^{n} dS_i e_i}{\sum_{i=1}^{n} e_i}$$

where $dS_i$ is a real number that represents the synonymous divergence of gene i, $e_i$ is the normalized transcript expression value of gene i, and n is the total number of genes analyzed[9]. TdNI and TdSI are weighted means of nonsynonymous and synonymous sequence divergence respectively. For each developmental stage, using the effective number of codons (ENC) measure[106] for 18,727 genes, we calculated the TCBI (Supplementary Fig. 2 and Supplementary Data 5):

$$TCBI = \frac{\sum_{i=1}^{n} ENC_i e_i}{\sum_{i=1}^{n} e_i}$$

where ENC is a real number that represents the codon usage bias of gene i, $e_i$ is the normalized transcript expression value of gene i, and n is the total number of genes analyzed[9]. A lower TCBI value corresponds to a transcriptome with higher codon usage bias, and vice versa. To calculate the statistical significance of TAI, TdNI, TdSI, and TCBI profiles, we used flat-line test implemented in the R package *myTAI* V0.9.3[40]. The relative expression of genes for a certain phylostratum (ps) and developmental stage (s) (Supplementary Figs. 6 and 7) were calculated as follows:

$$RE(ps)_s = \frac{\bar{f} - \bar{f}_{\min}}{\bar{f}_{\max} - \bar{f}_{\min}}$$

Where $\bar{f}$ is the mean normalized expression value of genes from phylostratum (ps) in the given stage, while $\bar{f}_{\min}$ and $\bar{f}_{\max}$ are the minimal and maximal mean normalized expression values of genes from the phylostratum (ps) across all stages[1]. Relative expression values for a certain phylostratum range from 1 in the developmental stage where the mean normalized expression value is the highest and 0 where the mean normalized expression value is the lowest.

## Statistics and reproducibility

The differences in embryo induction potentials between the wild type and the *drm1/drm2* double mutant were statistically evaluated using a two-sided Student's *t*-test. We performed three individual experiments of SE using the herein-described procedure, each conducted on 50–100 embryos per line (Supplementary Data 10). The effect size was determined by calculating Hedges' g, which is generally interpreted in the following way: g = 0.2 small, g = 0.5 medium, g = 0.8 large effect size[107]. The calculations were performed using the effect size V0.8.9 R package[108]. Other statistical procedures are described in their corresponding sections in "Methods".

## Reporting summary

Further information on research design is available in the Nature Portfolio Reporting Summary linked to this article.

## Data availability

All transcriptome data have been deposited in NCBI's Gene Expression Omnibus under accession number GSE234231 and are available at the following URL: https://www.ncbi.nlm.nih.gov/geo/query/acc.cgi?acc=GSE234231. Source data underlying the graphs presented in the main figures can be found in Supplementary Data 11. All other data are available in Figshare[109] at https://doi.org/10.6084/m9.figshare.28309805.v1 or from the corresponding authors on reasonable request.

## Code availability

The custom-made code used in this study is available on GitHub at https://github.com/bacillus-biofilms/biofilm-data-analysis and Zenodo[110] at https://doi.org/10.5281/zenodo.14718116.

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

## Acknowledgements
This work was supported by the City of Zagreb, the Croatian Science Foundation under the project IP-2016-06-5924, the Adris Foundation, and the European Regional Development Fund (KK.01.1.1.01.0009 DATACROSS) to T.D.-L.

## Author contributions
T.D.-L., D.L.-L., and N.M. initiated and conceptualized the study, D.L.-L., N.M., M.J., and A.I. collected the plant material and performed wet lab experiments, S.K., K.B.V., M.F., N.Č., A.T., N.K., M.D.-L., K.V., and T.D.-L. performed bioinformatic analyses, S.K., K.B.V., N.K., and T.D.-L. prepared the figures and tables for publication. S.K., D.L.-L., N.M., and T.D.-L. wrote the manuscript. All authors read and approved the manuscript.

## Competing interests
The method for induction of somatic embryogenesis in grapevine described in this work is in part covered by the Croatian patent (HRP20190444A2) invented by D.L-L. and N.M. and held by the Faculty of Science, University of Zagreb. All other authors declare no competing interests.
