## [Transparent Peer Review file · Communications Biology]

Developmental phylotranscriptomics in grapevine suggests an ancestral role of somatic embryogenesis

Corresponding Author: Professor Tomislav Domazet-Lošo

This manuscript has been previously reviewed at another journal. This document only contains information relating to versions considered at Communications Biology.

Version 0:

Reviewer comments:

Reviewer #1

(Remarks to the Author)

Q1: Too much content in the Introduction section to effectively locate the exact scientific question, innovations and implications raised in that section.

Q2: In the Introduction section, most of the references are dated and less relevant research has been done in recent years.

Q3: No indication of the accuracy of the transcriptome data.

Q4: Only data-rich transcriptome analyses were performed, lack of the functional validation of corresponding ideas.

Q5: The discussion section lacks a systematic summary of the findings.

Decision: Reject or supplement relevant trials before submitting.

Reviewer #2

(Remarks to the Author)

General comments:

Koska and co-authors developed the somatic embryogenesis system in grapevine (*Vitis vinifera*) and sequenced its developmental transcriptomes, found the heart stage expressed the most conserved transcriptome contrasted to the torpedo stage as most inert of Arabidopsis. The methods are well described, results are interesting and well developed.

Specific comments:

The molecular mechanisms reflected by the gene expression patterns in grapevine shall be discussed more insightful.

Reviewer #3

(Remarks to the Author)

The authors developed somatic embryogenesis in grapevine and sequenced its developmental transcriptomes, they combined the evolutionary properties of grapevine genes with their expression values from early induction until the formation of juvenile plants. Overall, this manuscript does not bring much data enough and new insight for the journal. There are several statements not supported with sufficient data. Although the work seems to have been carried out with care, I regret that the manuscript cannot undergo peer review at this time and, thus, cannot be further considered for publication.

1. Comment: The manuscript is just a general description of the results of RNA-SEQ sequencing data of grape somatic embryogenesis, lack of in-depth exploration of mechanism.

2. Comment: The introduction is too long, I suggest the author simplify and condense it.
3. Comment: Too many literatures are cited, and most of them are pretty old, while some important and recent literatures are not included.
4. Comment: Used too many paragraphs to describe the whole results of sequencing. But the key points of the article were rarely mentioned in Results Chapters.
5. Comment: -The discussion must be diligently thinking. To attract the attention of the journal audience the discussion must deep in the mechanisms underlying the process involved more than a comparison of the results with previous publications. Give practical ideas and applications for the target audience

Version 1:

Reviewer comments:

Reviewer #1

(Remarks to the Author)

The article is a systematic and effective work that is innovative and relevant.

Manuscript section: Good.

Subsidiary material section: Please add valid annotations for all subsidiary information, like the figures in the manuscript.

Reviewer #2

(Remarks to the Author)

The authors have addressed all major concerns raised in my review, making significant improvements to the manuscript. I recommend accepting this revised version.

Reviewer #3

(Remarks to the Author)

After reviewing the article on the development of somatic embryogenesis in grapevine, the authors have provided a detailed response to the reviewers' comments and have made corresponding revisions and additions to the manuscript. Upon careful consideration, I think this article holds significant academic value, particularly in offering new insights into the ancestral role of somatic embryogenesis. Therefore, I suggest that this article can be accepted. I believe it will have a positive impact on research in this field.

8 Nov 2024.

Point-by-point response to referees

Manuscript “*Somatic embryogenesis of grapevine (Vitis vinifera) expresses a transcriptomic hourglass*” by Koska *et al.* submitted to *Communications Biology*.

Reviewers' comments:

Reviewer #1 (Remarks to the Author):

Q1: Too much content in the Introduction section to effectively locate the exact scientific question, innovations and implications raised in that section.

We thank the reviewer for this remark. In response, we have streamlined the introduction to focus on our primary scientific question: Does somatic embryogenesis exhibit phylogeny-ontogeny correlations, and if so, what is the nature of these correlations?

Q2: In the Introduction section, most of the references are dated and less relevant research has been done in recent years.

We thank the reviewer for suggesting to refresh the references. We now carefully searched the literature and updated references with more recent work wherever possible.

Q3: No indication of the accuracy of the transcriptome data.

In the Supplementary Data 1, we now provide all details on the quality of transcriptome sequencing including mapping and coverage details.

Q4: Only data-rich transcriptome analyses were performed, lack of the functional validation of corresponding ideas.

We agree with the reviewer that our work primarily represents a phylotranscriptomic study. Similar to a recent phylotranscriptomic study on brown algae published in *Nature* (doi.org/10.1038/s41586-024-08059-8), we believe that functional validation, while beneficial, is not essential to substantiate our main conclusions on macroevolutionary imprints in somatic embryogenesis. Nonetheless, we have included a functional analysis of the key methylation gene *DRM2*, which we identified as highly expressed during the induction phase of somatic embryogenesis. Our results demonstrate that the loss of *DRM2* function significantly impairs somatic embryo induction (Figure 5).

Q5: The discussion section lacks a systematic summary of the findings.

We have now expanded the discussion by providing additional information on the significance of the evolutionary and functional findings presented in the paper.

Decision: Reject or supplement relevant trials before submitting.

Reviewer #2 (Remarks to the Author):

General comments:

Koska and co-authors developed the somatic embryogenesis system in grapevine (*Vitis vinifera*) and sequenced its developmental transcriptomes, found the heart stage expressed the most conserved transcriptome contrasted to the torpedo stage as most inert of *Arabidopsis*. The methods are well described, results are interesting and well developed.

We thank the reviewer for finding our study interesting.

Specific comments:

The molecular mechanisms reflected by the gene expression patterns in grapevine shall be discussed more insightful.

We thank the reviewer for this comment. We now discuss in more detail molecular mechanisms behind gene expression patterns. For instance, we now provide a loss-of-function data for DRM2 and discuss its significance for the somatic embryogenesis.

Reviewer #3 (Remarks to the Author):

The authors developed somatic embryogenesis in grapevine and sequenced its developmental transcriptomes, they combined the evolutionary properties of grapevine genes with their expression values from early induction until the formation of juvenile plants. Overall, this manuscript does not bring much data enough and new insight for the journal. There are several statements not supported with sufficient data. Although the work seems to have been carried out with care, I regret that the manuscript cannot undergo peer review at this time and, thus, cannot be further considered for publication.

We thank the reviewer for their thorough evaluation of our manuscript. However, we respectfully disagree with the assessment that our work does not contribute new data or insights. In support of our claim, we point to a very recent, methodologically highly similar phylotranscriptomic study on brown algae development published in *Nature* (doi.org/10.1038/s41586-024-08059-8), which cites a preprint of this manuscript. In this revised manuscript, we have also made a concerted effort to more clearly highlight the scientific advances that our study contributes.

1. Comment: The manuscript is just a general description of the results of RNA-SEQ sequencing data of grape somatic embryogenesis, lack of in-depth exploration of mechanism.

We appreciate the reviewer's comment and the opportunity to clarify. Our study presents an in-depth phylotranscriptomic analysis of somatic embryogenesis, integrating sophisticated evolutionary analyses that, combined with transcriptome data, reveal the macroevolutionary processes underlying somatic embryogenesis. In fact, in some aspects, our study incorporates more advanced methodologies and analyses than those employed in the very recent *Nature* publication on brown algae we mentioned above.

To further illustrate the potential applications of our data for exploring molecular mechanisms, we have also added a functional loss-of-function analysis of the methylation gene *DRM2*, demonstrating its impact on somatic embryo induction (Figure 5).

2. Comment: The introduction is too long, I suggest the author simplify and condense it.

We agree with the reviewer that introduction could be shortened for clarity. In this version we simplified and condensed the introduction.

3. Comment: Too many literatures are cited, and most of them are pretty old, while some important and recent literatures are not included.

We thank the reviewer for noticing this. We now updated the literature by including more recent literature.

4. Comment: Used too many paragraphs to describe the whole results of sequencing. But the key points of the article were rarely mentioned in Results Chapters.

We now stressed more clearly important parts in the results section.

5. Comment: -The discussion must be diligently thinking. To attract the attention of the journal audience the discussion must deep in the mechanisms underlying the process involved more than a comparison of the results with previous publications. Give practical ideas and applications for the target audience.

We thank the reviewer for this suggestion. We are confident that the revised discussion carefully presents our evolutionary and functional findings and places them in a broader context. We have now added a specific reference to the *Nature* publication on brown algae mentioned above.

REVIEWER COMMENTS

Reviewer #1 (Remarks to the Author):

The article is a systematic and effective work that is innovative and relevant.

Manuscript section: Good.

Subsidiary material section: Please add valid annotations for all subsidiary information, like the figures in the manuscript.

We thank the reviewer for recognizing the relevance and innovativeness of our work. We have added all necessary annotations to the supplementary material as requested.

Reviewer #2 (Remarks to the Author):

The authors have addressed all major concerns raised in my review, making significant improvements to the manuscript. I recommend accepting this revised version.

We appreciate the reviewer's positive feedback and recommendation for acceptance. We are grateful for valuable insights, which have helped improve the manuscript.

Reviewer #3 (Remarks to the Author):

After reviewing the article on the development of somatic embryogenesis in grapevine, the authors have provided a detailed response to the reviewers' comments and have made corresponding revisions and additions to the manuscript. Upon careful consideration, I think this article holds significant academic value, particularly in offering new insights into the ancestral role of somatic embryogenesis. Therefore, I suggest that this article can be accepted. I believe it will have a positive impact on research in this field.

We thank the reviewer for thoughtful comments and positive evaluation of our manuscript. We are grateful for the recognition that our work provides new insights into the ancestral role of somatic embryogenesis. We also greatly appreciate the support for the acceptance of the article and for acknowledging its value within the field.

The authors developed somatic embryogenesis in grapevine and sequenced its developmental transcriptomes, they combined the evolutionary properties of grapevine genes with their expression values from early induction until the formation of juvenile plants. Overall, this manuscript does not bring much data enough and new insight for the journal. There are several statements not supported with sufficient data. Although the work seems to have been carried out with care, I regret that the manuscript cannot undergo peer review at this time and, thus, cannot be further considered for publication.

1. **Comment:** The manuscript is just a general description of the results of RNA-SEQ sequencing data of grape somatic embryogenesis, lack of in-depth exploration of mechanism.
2. **Comment:** The introduction is too long, I suggest the author simplify and condense it.
3. **Comment:** Too many literatures are cited, and most of them are pretty old, while some important and recent literatures are not included.
4. **Comment:** Used too many paragraphs to describe the whole results of sequencing. But the key points of the article were rarely mentioned in Results Chapters.
5. **Comment:** -The discussion must be diligently thinking. To attract the attention of the journal audience the discussion must deep in the mechanisms underlying the process involved more than a comparison of the results with previous publications. Give practical ideas and applications for the target audience

The authors developed somatic embryogenesis in grapevine and sequenced its developmental transcriptomes, they combined the evolutionary properties of grapevine genes with their expression values from early induction until the formation of juvenile plants. Overall, this manuscript does not bring much data enough and new insight for the journal. There are several statements not supported with sufficient data. Although the work seems to have been carried out with care, I regret that the manuscript cannot undergo peer review at this time and, thus, cannot be further considered for publication.

1. **Comment:** The manuscript is just a general description of the results of RNA-SEQ sequencing data of grape somatic embryogenesis, lack of in-depth exploration of mechanism.
2. **Comment:** The introduction is too long, I suggest the author simplify and condense it.
3. **Comment:** Too many literatures are cited, and most of them are pretty old, while some important and recent literatures are not included.
4. **Comment:** Used too many paragraphs to describe the whole results of sequencing. But the key points of the article were rarely mentioned in Results Chapters.
5. **Comment:** -The discussion must be diligently thinking. To attract the attention of the journal audience the discussion must deep in the mechanisms underlying the process involved more than a comparison of the results with previous publications. Give practical ideas and applications for the target audience